# Manipulation of emergent vortices in swarms of magnetic rollers

Gašper Kokot [1] & Alexey Snezhko[1]

Active colloids are an emergent class of out-of-equilibrium materials demonstrating complex collective phases and tunable functionalities. Microscopic particles energized by external fields exhibit a plethora of fascinating collective phenomena, yet mechanisms of control and manipulation of active phases often remains lacking. Here we report the emergence of unconfined macroscopic vortices in a system of ferromagnetic rollers energized by a vertical alternating magnetic field and elucidate the complex nature of a magnetic roller-vortex interactions with inert scatterers. We demonstrate that active self-organized vortices have an ability to spontaneously switch the direction of rotation and move across the surface. We reveal the capability of certain non-active particles to pin the vortex and manipulate its dynamics. Building on our findings, we demonstrate the potential of magnetic roller vortices to effectively capture and transport inert particles at the microscale.

---

[1] Materials Science Division, Argonne National Laboratory, 9700 South Cass Avenue, Argonne, IL 60439, USA. Correspondence and requests for materials should be addressed to A.S. (email: snezhko@anl.gov)

Out of equilibrium ensembles of colloidal particles powered by external energy input often demonstrate remarkable level of complexity when driven out of equilibrium. Complex coherent motion and self-organization occur at a multitude of length scales from animal flocks[1] to bacterial swarms[2] and out-of-equilibrium (active) colloids[3–20]. A significant effort has been dedicated to identifying the basic underlying rules governing emergence of self-assembled phases and coherent collective motion in ensembles of active agents[3,21–23].

Active systems, composed of colloidal particles, transduce the energy stored in the environment or delivered by an external field into mechanical motion. They represent a convenient platform that allows investigating in detail the onset of coherent motion and self-organization in out-of-equilibrium multi-particle ensembles. It is largely because of their controllability, size, diverse range of tunable interactions and the absence of complex biochemical factors obscuring studies in biological systems. Despite their seeming simplicity colloidal systems exhibit a remarkable level of complexity when driven away from equilibrium by external magnetic or electric fields[14,18,24–27]. Recent advances in active colloidal studies have brought to light two new complementary experimental systems with activity stemming from particles' spontaneous rotations leading to self-propulsion in the presence of an interface (colloidal rollers)[28,29]. One type of rollers (termed Quincke rollers)[28] have been realized as an outcome of the spontaneous electro-rotation of a dielectric sphere in

a conductive fluid when exposed to a static (dc) electric field, while another type of rollers[29] rely on the spontaneous rotation of a ferromagnetic sphere in a uniaxial alternating magnetic field. Remarkably, the emergence of flocks and large-scale vortices has been reported in those systems[13,29] similar to flocking behavior observed in granular systems, bacterial suspensions and biofilaments-molecular motors mixtures[4,30,31]. The emergence of a spontaneous vortex in ensembles of self-propelling agents was mostly observed in geometrically constrained environments and it is believed to be an essential prerequisite for the emergence of vortical motion[13,31]. The important challenges remaining to be addressed are how emergent flocks and vortices interact with non-active (passive) particles and how to use those interactions to effectively control and direct the emergent dynamics in those active systems. The answers to those questions will lay the basis for new controls of active systems ranging from future multi-component active materials to crowd control, animal migration patterning, and control of artificial swarms of robots or drones. Thus far the research was mostly addressing the survival of the colloidal flocks in disordered environment[32], influence of the static obstacles on active particle transport characteristics[33,34] and onset of transitions to glass[35] or jamming[34].

In this paper, we report the emergence of unconfined macroscopic vortices in the system of ferromagnetic colloidal rollers and uncover complex non-trivial dynamics of self-organized magnetic roller vortices. We demonstrate that observed spontaneous self-organization is not a consequence of geometrical restrictions or

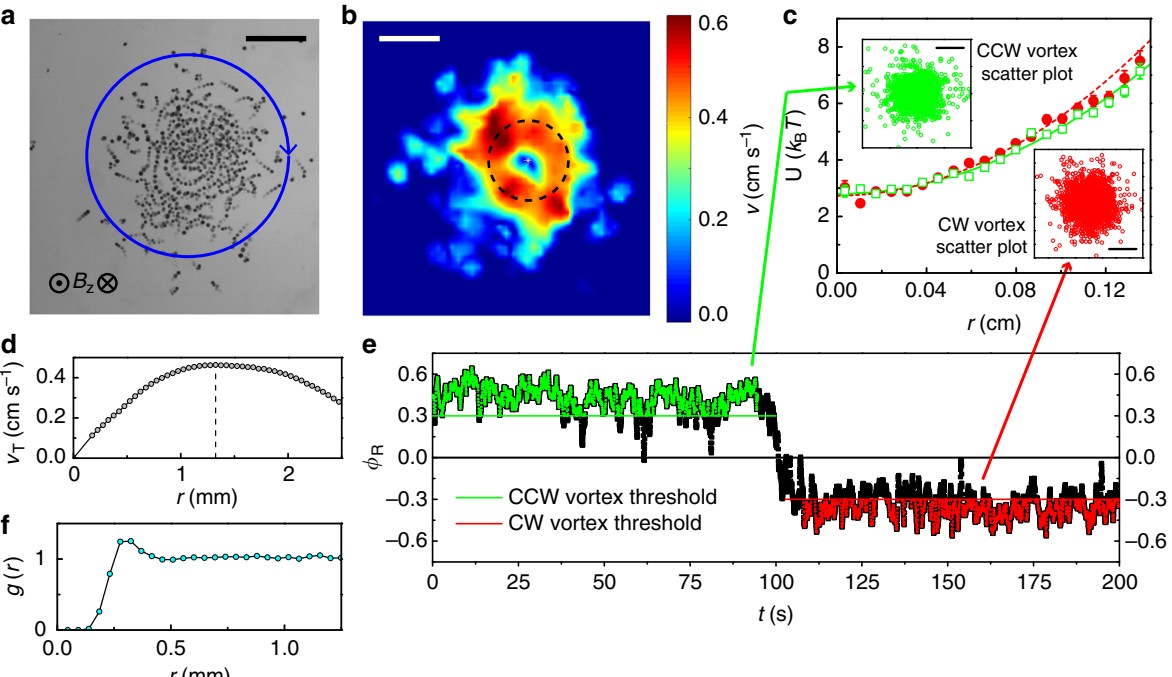

**Fig. 1** A magnetic roller vortex. **a** An overlay of five consecutive images showing a single roller vortex in a spherical potential well (radius of curvature 52 mm, rotating clockwise (CW). A particle grayscale intensity indicates time of the frame, light-colored particles being the initial frame and dark particles the final (see Supplementary Movie 1). Scale bar length is 2 mm. **b** Calculated velocity amplitude field. The white star symbol is the position of the vortex eye as determined by velocity and vorticity maps. Dashed circle around a vortex eye illustrates the core of the vortex as determined from the azimuthally averaged velocity profile of the vortex shown in **d**. Scale bar is 2 mm. **c** Potential energy $U$ of the vortex eye in the spherical well has a harmonic dependence on distance from center $r$. $U$ was calculated using Eq. 1 from scatter plots of the vortex eye positions (insets). Scale bars are 1. Error bars are the standard deviation of the measurements. As indicated by the order parameter $\phi_R$ in **e** the vortex spontaneously switched rotation direction during the course of the experiment. For scatter plots only frames with $|\phi_R| > 0.3$ were chosen. **d** Azimuthally averaged vortex tangential velocity profile versus distance from the center for the vortex shown in **b**. Dashed line marks the boundary of the defined vortex core where the rollers speed reaches maximum. **e** Time evolution of the polar order parameter $\phi_R$ in a roller vortex demonstrating a spontaneous switch of the vortex chirality. **f** The pair correlation function for the rollers forming the vortex. $g(r)$ has a single peak, implying the presence of a characteristic spacing between rollers. **a–e** The magnetic field direction is aligned with the gravity and oscillates in- and out-of-plane (amplitude $B_0 = 5.78$ mT, frequency $f_B = 40$ Hz)

finite system size and that a magnetic roller vortex has an ability to move across the surface. Remarkably, an active roller vortex can spontaneously switch the direction of rotation. We further use these self-assembled roller vortices to explore their complex interactions with inert scatterers and reveal the ability of passive particles with sizes above a certain limit to effectively trap the active vortex core and manipulate its dynamics. Manipulation of active unconfined vortices by passive pinning centers is reminiscent of a vortex matter control in superconductors by inclusions and defects[36]. We explore trapping potential of scatterers in detail and establish their pinning potentials with respect to active vortices. Moreover, we demonstrate the ability of a magnetic roller vortex to effectively capture and cage a mobile inert particle in the eye of the vortex and transport it across the interface. Our work provides new fundamental insights into behavior of a broad class of active systems where collective motion is caused by a fine interplay between rotational and translational degrees of freedom, and also suggest new techniques of control and transport of active colloids in general.

## Results

**Magnetic colloidal rollers**. In our experiments we sediment ferromagnetic nickel microspheres ($R_{Ni} = 69$ μm) dispersed in a liquid at the bottom of a glass container. A uniform vertical alternating magnetic field ($B_z = B_0 \sin(2\pi f_B t)$, where $t$ is time, $B_0$ is the field amplitude, and $f_B$ is the frequency) is used to energize the particles (see the Methods section). In a certain range of excitation field parameters[29] particles spontaneously break the symmetry of the clockwise/counterclockwise rotations experienced by a magnetic particle in a uniaxial field and start to spin[20,29] creating a directionally uncorrelated rolling motions of many particles. The particles steadily spin when the following condition is satisfied: $Im(\nu[-p^2, 2q]) - p > 0$[29]. Here the Mathieu characteristic exponent $\nu$ is the function of the two parameters $p = \alpha_r/(\omega I)$ and $q = \mu B_0/(\omega^2 I)$ where $\omega = 2\pi f_B$, $\eta$ is a fluid kinematic viscosity and $\mu$, $m$, $I = \frac{2}{5}mR_{Ni}^2$, $\alpha_r = 8\pi\eta R_{Ni}^3$ are correspondingly the magnetic moment, mass, moment of inertia and the rotational drag coefficient of a roller. In the state of a steady rotation the magnetic torque on the particle, particle, $T_m = \|\mu \times \mathbf{B}\|$, is

balanced by its viscous torque, $T_v \simeq \alpha_r \omega$, and characteristic viscous time $\tau_v$ is comparable with a time scale of the applied field frequency $\tau_f$. The frequency of the alternating magnetic field controls the speed of the rolling motion and it is employed to tune the activity in the system. Magnetic colloidal rollers demonstrate strong propensity toward the onset of a large scale collective motion. A set of dynamic phases ranging from gas to intermittent flocks and emergence of a global vortex has been observed in this system in response to changes in the rollers' activity[29]. The formation of vortices, see Fig. 1a, b and Supplementary Movie 1 and 2, is observed in our system in a relatively narrow region of the field excitation frequencies (about 10 band); however, a magnetic roller vortex is a robust reproducible entity. The chirality of the vortex rotation (clock- or counterclockwise) is random from experiment to experiment. The core of the vortex is characterized almost linear tangential velocity profile (see Fig. 1d). Remarkably, the emergence of the magnetic roller vortices does not rely on the presence of geometrical boundaries or confinement. A vortex was successfully observed in a flat bottom glass container with the surface area significantly larger than the area of the self-organized roller vortex so that the rollers forming the vortex do not interact with the container's walls (see Supplementary Movie 3).

To quantify the vortex we calculate the polar order parameter, $\phi_R(t) = \frac{1}{N}\sum_{i=1}^{N} \hat{\mathbf{e}}_{\varphi_i}(t) \cdot \hat{v}_i(t)$. Here, $\hat{\mathbf{e}}_{\varphi_i}$ are the in-plane unit vectors in angular direction, $\hat{v}_i$ are the in-plane velocities over the grid points and $N$ is the number of grid points. In the ideal global polar state the order parameter will reach unity in magnitude, but in the case of the roller vortex state in our system $\phi_R(t)$ is smaller since the vortex is finite (not global) and some of the particles are in the gas phase beyond the boundaries of the vortex. As the system is dynamic by nature the polar order parameter fluctuates around a mean value with a normal distribution of fluctuations (see Supplementary Note 1 and Supplementary Figure 1).

**Chirality state of a roller-vortex**. Rollers from the gas perpetually join and leave the vortex and the whole system is very dynamic. While the direction of the vortex rotation is randomly selected by

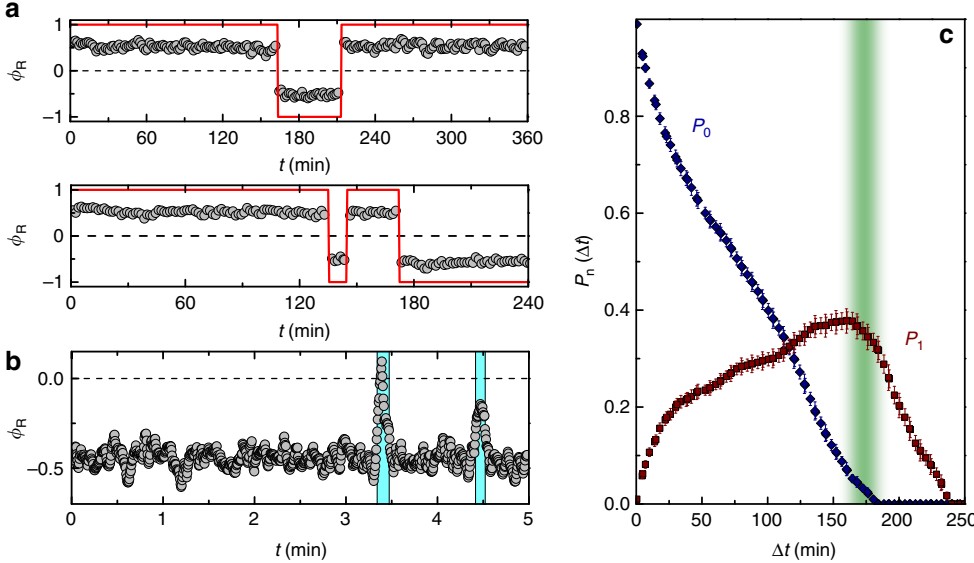

**Fig. 2** Chirality switching statistics of a roller vortex. **a** Chirality switching events illustrated by the polar order parameter of a roller vortex formed in a soft harmonic gravitational confinement. $B_0 = 5.78$ mT, $f_B = 40$ Hz. **b** Events of the flock intermittency not resulted in a roller vortex chirality switch.
**c** Probability distribution functions for the roller vortex to have no chirality switching events ($P_O$) or have one successful chirality change ($P_1$). Error bars are the standard deviation of the measurements

the system from experiment to experiment, occasionally the vortex can spontaneously change its chirality state as demonstrated in Fig. 1e. In Fig. 2a we demonstrate chirality switching events as they manifest in the polar order parameter of a roller vortex formed in a soft harmonic gravitational confinement. As one can see from the graph the vortex chirality switching is stochastic and relatively rare event on the scale of a typical experiment. The spontaneous chirality switching usually proceeds through the intermittent formation of flocks and takes of the order of 10 s (also see Supplementary Movie 4). During this intermittent process the rollers again randomly selects new chirality state that also can result in instances when vortex falls back to its previous chiral state as shown in Fig. 2b. To get an additional insight into the statistics of the vortex chirality switching in our system we extracted probability distribution functions (Fig. 2c) for the vortex to have no switching events ($P_0$) and have one successful chirality switching ($P_1$). Both curves suggest that on average in our system a vortex chirality switching event happens on the scale of about 170 min. As we showed previously in simulations[29] phase synchronization between rollers plays an essential role in the formation of the roller vortex phase and apparent intermittencies in the vortex behavior resulting in vortex chirality switching most probably stem from the temporary loss and subsequent recovery of the global phase synchrony of the rollers comprising the vortex. Temporary local loss of synchronization can be triggered in the experiments by the collisions with neighboring rollers that have pronounced shape imperfections resulting in abnormal rotational diffusion.

**Unconfined roller vortices**. In contrast to its Quincke roller counterpart where nearly all particles are concentrated along the container boundary in the vortex phase[13], the magnetic roller vortex core has almost a 'solid' structure. Its density is approximately uniform in the core (see Supplementary Movie 2). To quantify the structure of the vortex core the pair correlation function $g(r)$ has been calculated for the rollers forming the vortex (see Fig. 1d). It exhibits a pronounced peak at about twice the particle diameter indicative of a short-range spacing between rollers forming the vortex.

Spontaneous formation of the magnetic roller vortices at a flat surface is a nontrivial collective phenomenon. To form and maintain a roller vortex a certain local number density of rollers has to be met. In a harmonic gravitational confinement[29] this is automatically reached due to the herding of rollers by the confining potential. At a flat surface the situation is different, however the system is able to spontaneously form and maintain a local vortex. The number density of particles inside a roller vortex usually exceeds the average surface number density of the system (about 11 mm$^{-2}$ inside the vortex versus 6 mm$^{-2}$ overall in the system), see Supplementary Movie 3. Dynamic local densifications of rollers forming the vortices at a flat surface are driven by the rollers themselves and are intrinsic property of the magnetic roller system. The size of the vortex is selected by a dynamic self-induced densification and a range of different sizes can be realized in the system at the same experimental conditions, see Supplementary Note 2 and Supplementary Figure 2. It is possible to have vortices as small as 1.2 mm and as large as 3.1 mm in diameter. However, the large vortices are less stable and may on average fall apart faster to form smaller entities; on the other hand small vortices may evaporate to a gas. The most probable size in the studied system was about 2 mm. This size could be altered by the number density of rollers, however it is insensitive to the driving field frequency or amplitude manipulations. In a concave surface case the needed density for the vortex formation is maintained and stabilized by a soft confinement and much wide

range of vortex sizes can be observed. Once formed, the roller vortex is a dynamic entity and can move around the surface for minutes before it disintegrates due to interactions with obstacles or other flocks.

The main mechanism behind dynamic densifications forming the vortex in our system is analogous in nature to Vicsek flocks[37] but driven in our system by a fine interplay between flows (advection forces) generated by individual magnetic rollers and magnetic interactions between rollers. In particular, each magnetic roller has a complex time-averaged interaction profile: rotation of the sphere in the fluid (Re > 1 for rollers, inertia is important) creates attractive hydrodynamic interactions in the lateral direction (along the axis of rotation, perpendicular to the direction of the roller motion) and repulsive in the direction perpendicular to the axis of rotation (along the direction of the roller motion)[38,39]. As a result, rollers hydrodynamicaly attract neighbors laterally and repel them if they are along the rollers direction of motion. Both forces decay as $1/r^3$ with a distance[39,40]. Time averaged magnetic interactions, on the other hand, are attractive along the rollers direction of motion and repulsive in lateral direction. Corresponding magnetic time averaged forces decay as $1/r^4$[15]. Thus, hydrodynamic and magnetic interactions between rollers keep the roller vortex from falling apart (long range attractions prevent rollers from departing the vortex), or collapsing to clusters (short range repulsions keep rollers from getting to close to form chains or clusters). However, collisions with obstacles (scatterers), or other flocks of rollers may disturb the balance and phase synchrony[29] of the rollers forming the vortex and it may get transformed to flocks or disintegrate.

To demonstrate the pivotal role of the induced hydrodynamic interactions in the emergence of vortices in a magnetic roller ensemble we completely eliminated the hydrodynamic effects by performing similar experiments in air. This is possible since spontaneous rotation of magnetic rollers do not rely on the presence of a liquid (in contrast to Quincke rollers). No vortices or flocks have been observed in a full range of the field parameters and number densities, where steady rolling of particles was possible in air, also see Supplementary Movie 5.

**Roller vortex and inert scatterers**. Once formed, the roller vortex is not a static entity. The vortex core explores the bottom surface of the experimental container (see Supplementary Movie 3), and it can be considered itself as a quasi-particle. To investigate in detail the effects of inert scatterers on dynamics of a single magnetic roller vortex we isolated the vortex in a soft harmonic gravitational well realized by a container with a concave bottom (we use a glass lens with a radius of curvature $R_{curv}$ = 52 mm). This way the vortex gets gravitationally localized around the center of the lens that prevents it from escaping the field of view or moving too close to the walls of the container that may disrupt or modify the vortex under investigation.

The roller vortex explores the vicinity of the concave surface center. We determined the position of the vortex eye (where velocity is close to zero, see Fig. 1b) for each time step and treat it as a quasi-particle exploring the potential landscape. The confining potential imposed by the concave surface on the vortex could be estimated using inverted Boltzmann equation[41],

$$U(r)/(k_B T) = -\ln[n(r)/A]. \tag{1}$$

Here $n(r)$ is a radial displacement histogram, and $A$ is a constant dependent on total number of measurements. Our observations (see Fig. 1c) reveal a harmonic displacement dependance of the potential $U = \frac{1}{2}k_p r^2$, where $k_p$ is the effective potential stiffness felt by the active vortex on the concave surface.

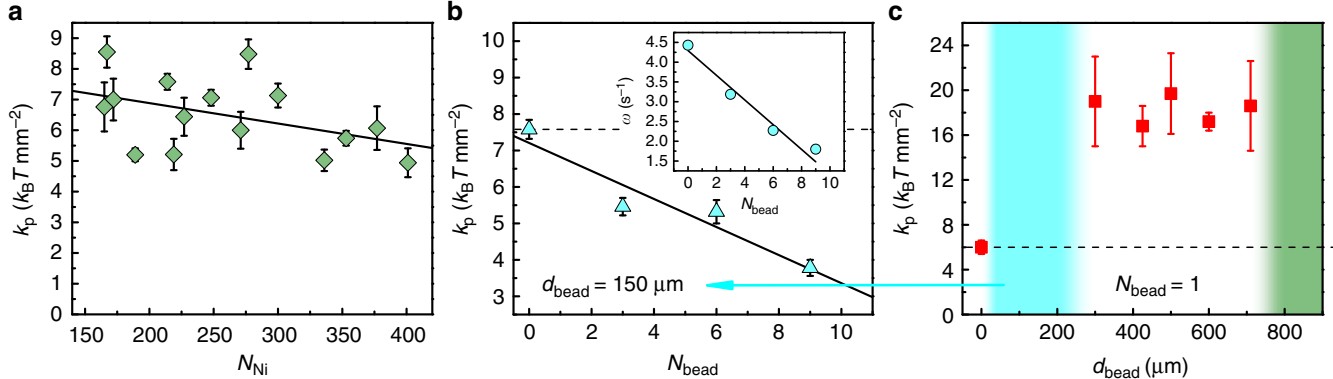

**Fig. 3** Potential stiffness manipulation with passive scatterers. **a** Trapping potential stiffness as a function of number of magnetic rollers comprising the vortex $N_{Ni}$. The number of rollers has a weak influence on the trapping potential. Black solid line is a linear fit. $B_0 = 5.78$ mT, $f_B = 40$ Hz. **b** Effect of small scatterers ($d_{bead} = 150 \pm 9\ \mu m$) on the vortex trapping potential $k_p$ (blue triangles, dashed line marks the value for a pristine vortex). Vortex angular speed $\omega$ (inset) decreases with the scatterers number. Black solid lines are linear fits. $B_0 = 5.78$ mT, $f_B = 40$ Hz, $N_{Ni} = 214$. **c** Pinning of the roller vortex by intermediate bead sizes. Three-fold increase in stiffness of the trapping potential $k_p$ for intermediate sized scatterers (red squares, dashed line marks the value for a pristine vortex). The blue region indicates the system response as in **b** and the green area shows the bead sizes, where even a single scatterer destroys the vortex state. $B_0 = 7.54$ mT, $f_B = 43$ Hz, $N_{Ni} = 271$. Error bars in all panels are the standard deviation of the measurements

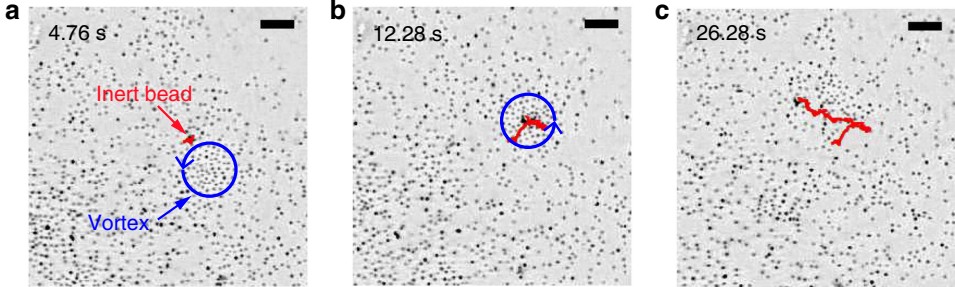

**Fig. 4** Inert bead capture and transport by a roller vortex. **a–c** The snapshots demonstrate the capture and transport of a passive glass bead ($d_{bead} = 500 \pm 25\ \mu m$). See also Supplementary Movie 2. The red curve marks the path traveled by the passive bead and the blue circle encloses the particles that are a part of the vortex. The arrow denotes the direction of the vortex rotation. **b** the scatterer is captured by the vortex through intermittent flocking. The passive particle is transported by the vortex for about 14 s. The vortex spontaneously disintegrates (frame **c**) to release the bead. The energizing field: $B_0 = 6.07$ mT, $f_B = 43$ Hz. The scale bars are 2 mm

The extracted potential stiffness shows only very weak dependence on the size of the roller vortex, as demonstrated in Fig. 3a, and an active vortex almost twice the size (in number of rollers involved $N_{Ni}$) remains approximately as mobile as the smallest one. Indeed, the vortex speed on a flat surface does not show dependence on the size, see Supplementary Figure 2.

To explore the behavior of the magnetic roller vortex in the presence of passive scatterers we added a small amount of nonmagnetic glass spheres (see Methods). Scatterers with diameter less than about 300 μm get dispersed and incorporated inside of the rotating body of the roller vortex. At first glance the roller vortex does not 'see' them. Nevertheless, since rollers collide with scatterers they consume part of the vortex energy resulting in overall decrease of the effective vortex angular speed $\omega$ with the number of scatterers $N_{bead}$, see inset of Fig. 3b. Surprisingly, the stiffness of the confining potential felt by the vortex decreases (Fig. 3b), as if the core of the vortex becomes suddenly more mobile with the number of scatterers. The contradiction is resolved by the fact that the increase in the number of scatterers leads to the instability of the vortex structure so that it adjusts its position by spontaneous re-assembly of itself in a new location rather than by a continuous motion. This leads to apparent sudden vortex core shifts and results in an effective softening of the observed potential stiffness. Continuous increase of the scatterers' number eventually jeopardizes the existence of the

vortex state. Alternatively the vortex state can be destroyed by a single large particle ($d_{bead} > 710\ \mu m$, green region in Fig. 3c).

Intermediate size beads ($300\ \mu m \leq d_{bead} \leq 710\ \mu m$) interact with the roller vortex in a significantly different manner. These beads get pushed inside of the eye of the vortex core. Once the bead is in the eye, the roller vortex mobility is suppressed and it gets pinned by the inert bead. This is revealed by almost three-fold increase in the effective stiffness of the confining potential $k_p$ shown in Fig. 3c for a range of inert particles sizes. The stronger confining potential in the presence of those inert beads in the vortex eye is due to the scattering of the rollers close to the vortex eye on the bead. As all rollers are hydrodynamically coupled and synchronized[29] within the vortex, collisions will keep a vortex eye on the bead to minimize obstructed motion of the rollers and sustain the vortex. This observed behavior is similar to pinning of Abrikosov vortices in superconductors by defects and inclusions used to improve transport properties of superconductors[36]. The property of the magnetic roller vortices to be pinned by certain inert defects provides us with a new tool to manipulate vortex dynamics in active roller suspensions. Conversely, if the inert particle is mobile the vortex can capture it inside of the core and transport with the roller vortex motion. This scenario is demonstrated in Fig. 4b where the inert 500 μm particle has been captured and transported for about 14 by the magnetic roller vortex generated at a flat surface, see Fig. 4 and Supplementary

Movie 6. Interestingly, the capture of the passive particle proceeded through intermittent flocking of the vortex' rollers around it. In a related approach[42] the trapping and manipulation of microscopic objects has been realized with the help of an induced hydrodynamic vortex generated by a rotating magnetic micro-wire in a rotational magnetic field. There the trapping force is very local (a few microns) and directional (the trap should approach at a specific angle to the object). In contrast, while each individual roller creates a similar hydrodynamic microvortex and is capable of hydrodynamic trapping of microparticles, the self-assembled roller vortex (collection of cooperating rollers) has a significantly extended trapping range (up to a few millimeters) and allows caging and manipulation of much larger particles compared to a single roller.

## Discussion

Our results demonstrate that active vortex patterns do not always require geometrical confinement for their observation and stem from the interplay between self-propulsion and particle alignment interactions. We observed multiple magnetic roller vortices in experimental cells with surface area much larger than a characteristic vortex size. A formation of a global vortex state has been achieved in a gravitationally confined (convex bottom) containers. We demonstrate that a self-organized roller vortex has an ability to move across a surface and spontaneously switch the direction of rotation. The main advantage of our model system compared to other realizations of active matter is that all interactions in the particle ensemble are well characterized and could be easily tuned by the parameters of the alternating magnetic field. Besides, the mechanism of rolling is liquid independent. Our work also sheds light on complex relationships between the roller vortices and inert scatterers. We have revealed that inert scatterers can be used as an effective tool to manipulate active vortex dynamics. We show that a roller vortex can be pinned by a inert obstacle of a certain size and conversely an active vortex has a potential for caging and transporting inert mobile objects. Building upon our findings, we demonstrated successful capture and diffusive transport of an inert bead by a self-assembled magnetic roller vortex. The ability to manipulate active colloidal structures is crucial for the development of directed transport at the micro-scale and progress of self-assembled micro-robotics. Our work suggests new approaches and techniques for manipulation of active colloidal materials.

## Methods

**Experimental details**. The experiments were performed cylindrical glass containers 5 cm in diameter with a flat or concave shaped bottom. We employed an optical lens with radius of curvature 52 to create a concave bottom. Ferromagnetic Ni microparticles (Alfa Aesar) with an average radius of $R_{Ni} \sim 69$ μm (62−75 μm uniform size distribution), density $\rho = 8.9$ g cm$^{-3}$ and average magnetic moment $\mu \simeq 2 \cdot 10^{-5}$ emu were dispersed in the isopropanol (kynematic viscosity $\eta = 2.5$ cSt) or water (kynematic viscosity $\eta = 1$ cSt). Re $\simeq 2.5$ in water. The system is energized by a pair of custom made precision Helmholtz coils. Uniaxial alternating magnetic field $(B_z = B_0 \sin(2\pi f_B t))$ is applied collinear with gravity and facilitates random rolling of Ni microparticles on the bottom of the container. Amplitude of the magnetic field was in the range 5−8 mT. Non-magnetic glass microparticles (Novum Glass LLC: U-710, U-600, U-500, U-425, U-300, U-150, and U-45) were used as inert scatterers with diameters in the range 45−710 μm. Dynamics of active microrollers is captured with a fast CMOS camera (IDT) mounted on a microscope stage.

**Relevant timescales**. Viscous time is $\tau_v \sim L^2/\eta = 2 \cdot 10^{-2}$ s, here $L$ is a typical size of the magnetic particles ($2^*R_{Ni} \sim 140$ μm). External alternating magnetic field: $\tau_f \sim 1/f = 2.3 \cdot 10^{-2}$ s.

**Data analysis**. Image processing, particle image velocimetry and data analysis were performed using custom scripts, ImageJ and MatPIV package for Matlab. The position of the vortex eye was determined from vorticity and velocity maps. About 24 h of the experimental video data capturing roller vortex dynamics have been used for determination of the probability distribution functions of the chirality

switching events for the roller vortex confined in a harmonic gravitational well. The scatter plots used to determine a radial displacement histogram $n(r)$ contained more than 2000 points for each vortex realization.

**Data availability**. The data in support of the reported findings are available from the corresponding author upon request (snezhko@anl.gov).

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

## Acknowledgements

We thank Dr. James E Martin for use of his large volume Helmholtz setup for test experiments on vortex states in large surface area containers. The research was supported by the U.S. Department of Energy, Office of Science, Basic Energy Sciences, Materials Sciences, and Engineering Division.

## Author contributions

A.S. conceived the research. G.K. and A.S. carried out the experiments and analyzed the data. All authors wrote the manuscript.

## Additional information

**Competing interests:** The authors declare no competing interests.

