## [Peer Review File · Nature Communications]

Reviewers' comments:

Reviewer #1 (Remarks to the Author):

The article by Kokot and Snezhko is an experimental work investigating the dynamics of driven magnetic colloidal rollers that are energized by a perpendicularly applied oscillating magnetic field.

The authors observe the formation of filled vortices that are unconfined, in contrast to their previous findings reported in Science advances.

Furthermore, the authors demonstrate that such vortices have an exciting dynamics, being able to switch randomly the sense of rotation, being pinned by large non magnetic colloids introduced in the dispersing medium.

The latter attempt resembles pinning in high Tc superconductors, and may provide a useful way to manipulate driven and active systems that displaying such behavior.

Thus the article will for sure inspire the work of different theoretical groups interested in understanding the dynamics in such complex environments.

The article is very well written, easy to read and understood, and the subject rather timely. Thus I find it of broad reach and well suited for the chosen journal, Nature comm.

I have some comments/suggestions, that

the authors are willing to address:

1) Are the reported structures vortices or mill patterns?

How the azimuthal velocity varies along the radial direction?

It is indeed reported in Fig.1b, but maybe a 1D

averaged over circular shells will also help to clarify.

2) The authors uses the $g(r)$ to extract the pair potential

between the particles in a vortex.

It is possible to use another approach?

This technique is in general used for thermodynamic equilibrium

cases (or near equilibrium ones), where thermal fluctuations

and Boltzmann statistics is justified.

Indeed from the $g(r)$ one can get the

Ornstein-Zernike

relationship and then $u(r)$.

In the case of the authors probably

this assumption is not valid, given the strongly out of equilibrium

system, where the balance between hydrodynamic and magnetism

dominates over $k_B T$.

3) The spontaneous switching of the vortex chirality

is an intriguing effect. It is possible to provide more statistics in this sense,

to see whatever the phenomenon is effectively random?

The order parameter show strong fluctuations around its mean,

how the distribution looks like?

4) A similar work, but not based on dynamic self-assembly

and that demonstrates mobile micro-vortices generated by an individual nanorod is:

Nano Lett., 2012, 12, 156–160.

The authors could comment in the text the differences and advantages of their approach.

5) Others related works on individual and collective

magnetic rollers the authors could cite are:

- Phys. Rev. Lett. 115, 138301 (2015)

- Phys. Rev. Applied 3, 051003 (2015).

Reviewer #2 (Remarks to the Author):

Manipulation of emergent vortices in swarms of magnetic rollers

Gasper Kokot and Alexey Snezhko

In this manuscript, the authors investigate collective phenomena in a system composed of rotating ferromagnetic colloids near a surface. They apply a 1D alternating magnetic field in a direction perpendicular to the surface, and the magnetic colloids roll on the surface due to the ensuing hydrodynamic translation-rotation coupling. The rollers exhibit collective rotational motion due to the interactions among the particles.

The work is an extension of their recent work (Kaiser et al., 2017; Science Advances), and I can identify two major differences from the previous work. Firstly, they observe that the collective rotation, which was reported in the previous paper with a concave surface, can be observed even for a flat surface. Secondly, they report interesting interactions between magnetic particle vortices with passive particles (scatterers).

The paper is easy to follow, and the observations have sufficient novelty and impact. However, the paper does not provide sufficiently strong theoretical perspective to explain the phenomenon. As it stands, it does not make a convincing story, and there are a number of points that need to be addressed.

Major Comments:

1. It is surprising that the roller vortices can be observed even for a flat bottom container. At the same time, it is difficult to understand why the vortex state stays for a long time without breaking. The physical origin of the collective motion needs to be explained and clarified.

2. What selects the size of the vortex in the case of a flat surface? How can the vortex size be different from a concave surface case?

3. Is there any way to change the size of the vortex in the flat surface case? Is it a function of the strength/frequency of the external field?

4. In Figure 2(c): what is the mechanism behind the observed stronger confining potential k_p in the presence of scatterers?

Minor Comments:

1. Some important parameters are not mentioned in the paper: viscosity/density of the solution, size of the whole container, magnetic moment of the particle, and density of each particle.

2. In Figure 2: Why is there no k_p value in the blue region? It is interesting to report the transition of k_p value in a range $d_{\text{bead}} = 0$ to 300.

Reviewer #3 (Remarks to the Author):

Kokot and Snezhko describe the formation of localized vortices within ensembles of active colloidal rollers powered by oscillating magnetic fields. These vortices are distinct from those described

previously for Quincke rollers in that they emerge spontaneously (under certain conditions) in the absence of geometric confinement and are characterized by a uniform particle density throughout the vortex area. The Authors show how roller vortices can be confined within a gravitational potential well and investigate their interactions with passive particles. Larger passive particles were observed to localize in the eye of the vortex. In this way, a roller vortex was capable of capturing and transporting passive cargo along the surface.

These vortex structures are fascinating and quite unlike related phenomena observed previously. I would be pleased to see this manuscript published in Nature Communications. However, the Authors should do more to explain the key physics underlying the rollers and their organization into vortices. In particular, the following items are not discussed in sufficient detail (if at they are discussed at all).

1) What sets the frequency at which vortices form? What is the relevant physics? For example, one might guess that particle motions arise through competition of magnetic and viscous torques at low Reynolds numbers. These two torques are of comparable magnitude on a particular time scale. How does that time scale compare to that of the applied frequency? On a related note, what is the magnetic moment of the particles? Perhaps these questions were addressed in previous work (ref. 26); however, it is necessary to repeat here for the benefit of the reader.

2) How do particles interact? Is it primarily magnetic dipole-dipole interactions or hydrodynamic interactions? How do these interactions facilitate vortex formation? Based on the “solid”-like character of the vortices, one might conjecture that repulsive dipolar interactions are significant.

3) How does the speed of the vortex scale with its size? The Authors already have data on vortices of different sizes (Fig. 2a).

4) What sets the characteristic size of the vortices (assuming there is a preferred size)?

Reply to Reviewer 1

We thank the Reviewer for a careful reading of our work and comments aimed to improve the clarity of our presentation. We are glad that the Reviewer finds our work well written, timely and consider it appropriate for publication in Nature Communications. The Reviewer made a number of comments and suggestions, which we fully addressed in the revised manuscript. Below are point-by-point replies to the Reviewer's comments:

1) *Are the reported structures vortices or mill patterns? How the azimuthal velocity varies along the radial direction? It is indeed reported in Fig.1b, but maybe a 1D averaged over circular shells will also help to clarify.*

The reported active rollers vortices are indeed vortex like patterns. To better demonstrate the appearance of the emergent vortices we included an additional slow-motion video of the vortex pattern where vortical motion of rollers is clearly seen (new Supplementary Video 1). We also included now the data on azimuthally averaged velocity of the rollers inside of the vortex. It is now supplied as a new panel in updated Figure 1. The core of the vortex is characterized by almost linear velocity profile. A corresponding discussion has been added to the manuscript.

2) *The authors uses the $g(r)$ to extract the pair potential between the particles in a vortex. It is possible to use another approach? This technique is in general used for thermodynamic equilibrium cases (or near equilibrium ones), where thermal fluctuations and Boltzmann statistics is justified. Indeed from the $g(r)$ one can get the Ornstein-Zernike relationship and then $u(r)$. In the case of the authors probably this assumption is not valid, given the strongly out of equilibrium system, where the balance between hydrodynamic and magnetism dominates over kBT .*

In the manuscript we only use the radial distribution function $g(r)$ to demonstrate a presence of a short range spatial order (illustrated by a clear first peak in the distribution $g(r)$). This short range characteristic spacing gives a “solid-like” look of the vortex core. This is the only intended use of $g(r)$ in the manuscript.

We agree with the Reviewer that use of $g(r)$ for extracting interaction potential *between particles* would not be justified, and as such we did not do it. Instead, we consider a whole vortex as a (quasi) particle. This new “particle” has certain energy and explores harmonic gravitational potential energy landscape imposed by the use of a concave surface of the container. And for this situation we can use Boltzmann statistics to extract effective potential $U(r)$ felt by the vortex from the displacement histogram of the vortex eye.

We modified the text to better explain the procedures.

3) *The spontaneous switching of the vortex chirality is an intriguing effect. It is possible to provide more statistics in this sense, to see whatever the phenomenon is effectively random?*

Indeed, the spontaneous chirality switching phenomenon is intriguing. Vortex chirality switching is a stochastic effect and proceeds through intermittent formation of flocks. During that flock intermittency that lasts of the order of 10 sec the system randomly selects new chirality state and thus one can also have instances when chirality jumps back to the original state. In the revised manuscript we introduced an additional Figure (now Figure 2) addressing the vortex chirality switching phenomenon. New figure demonstrates a few examples of the polar order parameter behavior during the successful switching (Fig. 2a) and instances when system after flock intermittency recovers the same chirality (Fig. 2b) of the vortex. The switching events are

relatively rare and a single vortex in harmonic confinement may stay in the same chirality state for hours. We collected more than 24 hours of additional data to analyze the statistics of chirality switching and now show it as panel C in the new Fig.2. There we plot two probability distribution functions: i) to find a vortex with no chirality switching events (P0) and ii) to have one chirality switching event (P1). Both curves suggest that on average a chirality switching event happens on the scale of about 170 minutes.

We added a new paragraph to the text of the revised manuscript describing the rollers vortex chirality switching, added new Figure 2 and Supplementary Video 4 illustrating the process of the vortex chirality switching through flocks intermittency.

The order parameter show strong fluctuations around its mean, how the distribution looks like?

We plotted the distribution of the polar order parameter fluctuations around the mean for the vortex. It shows normal (Gaussian) distribution. We added this information as a Supplementary Figure S1 and incorporated the corresponding comment to the manuscript.

4) *A similar work, but not based on dynamic self-assembly and that demonstrates mobile micro-vortices generated by an individual nanorod is: Nano Lett., 2012, 12, 156–160.*

The authors could comment in the text the differences and advantages of their approach.

We thank the Reviewer for bringing to our attention the paper that is indeed related and now is cited in the manuscript. In that paper authors used hydrodynamic micro vortex generated by a micro-wire in a rotating magnetic field. The technique is capable of trapping micro-objects by the hydrodynamic field and transporting it together with the wire using magnetic fields. In our case the trapping of the bead is driven by a collective phenomenon - ensemble of rollers spontaneously creating vortex pattern - rather than a single particle rotation. The cargo-bead is expelled towards the center of the roller vortex by the interactions with rollers and transported by the motion of the vortex pattern with the bead trapped in the “eye” of the vortex. The later technique is long-range (up to few millimeters - of the order of the roller vortex pattern size) while rotating wire trapping is local (the bead needs to be in the close proximity (microns) to the rotating trap) and very directional (the trapping wire should arrive only tangentially to the side of the bead for a successfully trapping event). We added a corresponding short discussion in the revised manuscript. In particular, we added the following:

“In a related approach [42] the trapping and manipulation of microscopic objects has been realized with the help of an induced hydrodynamic vortex generated by a rotating magnetic micro-wire in a rotational magnetic field. In that approach the trapping force is very local (a few microns) and directional (the trap should approach at a specific angle to the object). In contrast, while each individual roller creates a similar hydrodynamic microvortex and is capable of hydrodynamic trapping of microparticles, the self-assembled rollers vortex (collection of cooperating rollers) has a significantly extended trapping range (up to a few millimeters) and allows caging and manipulation of much large particles compared to a single roller.”

5) *Others related works on individual and collective magnetic rollers the authors could cite are: - Phys. Rev. Lett. 115, 138301 (2015); - Phys. Rev. Applied 3, 051003 (2015).*

We added the suggested citations.

Reply to Reviewer 2

We thank the Reviewer for the careful reading of our work and comments aimed to improve the clarity of our presentation. We are glad that the Reviewer finds that our work “have sufficient novelty and impact”. The Reviewer made a number of valid comments and suggestions mainly aimed to providing additional insights and theoretical perspectives to the observed phenomenon. In the revised manuscript we fully addressed all points raised by the Reviewer. We significantly extended the explanatory part and provided insights to the observed novel phenomenon. Below are point-by-point replies to the Reviewer's comments:

1) *It is surprising that the roller vortices can be observed even for a flat bottom container. At the same time, it is difficult to understand why the vortex state stays for a long time without breaking. The physical origin of the collective motion needs to be explained and clarified.*

Indeed, we agree with the Reviewer that formation of vortices at a flat bottom container (no soft gravitational confinement) is a nontrivial phenomenon. As we showed previously for the roller vortices formed in a soft gravitational confinement [29] the collective dynamics of the rollers is governed by the interplay of hydrodynamic and magnetic interactions resulting in an emergence of correlated motion in a certain range of excitation parameters (field amplitude and frequency). The vortex state corresponds to the maximum correlation length between rollers and exists close to the stability limit of the robust particle rotations [29]: $\text{Im}(\nu[-p^2, 2q]) - p > 0$. Here $\nu[x,y]$ is the Mathieu characteristic exponent function, $p = a_r/(\omega I)$, $q = \mu B_0 / (\omega^2 I)$, $a_r = 8\pi\eta R^3$ is the rotational drag coefficient, I is a momentum of particle inertia). Also as we previously demonstrated rollers forming the vortex are phase synchronized [29].

To form and maintain a roller vortex a certain number density of rollers needs to be met (as well as activity controlled by the parameters of the field [29]). In a harmonic gravitational confinement realized in Ref.[29] this is automatically reached due to the herding of rollers by the confining harmonic potential.

At a flat surface the situation is different. However, as we observe in the experiments at a flat surface the system still can spontaneously form and maintain a vortex. The number density of particles inside a roller vortex usually exceeds the average surface number density of the system (about 11 per mm^2 inside the vortex versus 6 per mm^2 overall in the system). We added a new Supplementary Video 3 of the roller vortex at a flat surface clearly demonstrating the unconfined roller vortex and density differences. Those self-generated sporadic local densifications forming the vortices at a flat surface are driven by the rollers themselves and are an intrinsic property of the magnetic roller system. Once formed the roller vortex is a dynamic entity and can move around the surface for minutes before it disintegrates due to interactions with obstacles or other flocks.

We believe that the main mechanism behind temporal densifications forming the vortex in our system is analogous in nature to Vicsek flocks in the model of self-propelled particles but driven in our system by a fine interplay of flows (advection forces) generated by individual magnetic rollers and magnetic interactions between rollers. In particular, each magnetic roller has a complex and anisotropic time-averaged interaction profile: rotation of the sphere in the fluid ($Re > 1$ for rollers, inertia is important) creates attractive hydrodynamic interactions in the lateral direction (along the axis of rotation, perpendicular to the direction of the roller motion) and repulsive in the direction perpendicular to the axis of rotation (along the direction of the roller motion) [38, 39]. As a result, rollers hydrodynamically attract neighbors laterally and repel them

if they are along the rollers direction of motion. Both forces decay as $1/r^3$ [39, 40]. Time averaged magnetic interactions, on the other hand, are attractive along the rollers direction of motion and repulsive in lateral direction. Corresponding magnetic time averaged forces decay as $1/r^4$ [15]. Thus, hydrodynamic and magnetic interactions between rollers keep the roller vortex from falling apart (long range attractions prevent rollers from departing the vortex), or collapsing to clusters (short range repulsions keep rollers from getting too close to form chains or clusters). However, collisions with obstacles (scatterers), or other flocks of rollers may disturb the balance and phase synchrony of the rollers forming the vortex and it may get transformed to flocks or disintegrate. To get detailed and quantitative theoretical insights into the observed unconfined roller vortex formation a full 3d hydrodynamics for arbitrary Re numbers needs to be solved (simulated) for rollers, that is computationally very demanding and as of now is not available (Stokes approximation used in [29] for harmonically confined rollers does not reproduce the observed phenomenon at a flat surface).

To further demonstrate the pivotal role of the hydrodynamic interactions in the formation of such vortices in a magnetic roller ensemble we completely eliminated the hydrodynamic effects in the experiment by conducting similar experiments in air. This is possible since spontaneous rotation of magnetic rollers do not rely on the presence of a liquid (in contrast to Quincke rollers). We added an additional Supplementary video demonstrating the state of the roller system in air (new Supplementary Video 5). No vortices or flocks have been observed in a full range of the number densities and field parameters where steady rolling of particles was possible in the air.

We incorporated corresponding discussions in the text of the manuscript.

2) What selects the size of the vortex in the case of a flat surface? How can the vortex size be different from a concave surface case?

The size of the vortex is selected by a dynamic self-induced densification and a range of different sizes can be realized in the system. We added a supplementary Figure S2 demonstrating a histogram of the observed roller vortex sizes realized on a flat surface at the same experimental conditions. It is possible to have vortices as small as 1 mm and as large as 3.2 mm in diameter. However, large vortices are less stable and may fall apart on average faster to form smaller entities; on the other hand small vortices may evaporate to a gas. On average the most probable size in the studied system was about 2 mm. This size could be altered by the number density of rollers.

In a concave surface case the needed density for the vortex formation is maintained and stabilized by a soft harmonic confinement and much wide range of vortex sizes can be observed. We added the corresponding discussion in the text of the manuscript.

3) Is there any way to change the size of the vortex in the flat surface case? Is it a function of the strength/frequency of the external field?

A system of rollers at a flat surface can support a wide range of vortex sizes at fixed experimental condition (see the discussion for question 2). A formation of the vortices happens in a narrow band of the frequency (about 10 Hz) and within that band change in the frequency does not affect the sizes of the observed roller vortices. Also increase in the field amplitude within the roller phase does not affect the sizes of the observed vortices.

We added corresponding comments in the revised manuscript.

4) *In Figure 2(c): what is the mechanism behind the observed stronger confining potential k_p in the presence of scatterers?*

The stronger confining potential in the presence of certain large scatterers in the vortex eye is due to the scattering of the rollers close to the eye on that particle. As all rollers are hydrodynamically coupled and synchronized within the vortex, collisions will keep a vortex eye on the bead (bead in the vortex eye) to support an unobstructed motion of the rollers in order to sustain the vortex. We added the discussion in the revised text.

Minor comments:

1. *Some important parameters are not mentioned in the paper: viscosity/density of the solution, size of the whole container, magnetic moment of the particle, and density of each particle.*

We added corresponding information in the Methods section of the manuscript.

2. *In Figure 2: Why is there no k_p value in the blue region? It is interesting to report the transition of k_p value in a range $d_{\text{bead}} = 0$ to 300.*

The k_p values in the blue region in fig 2c (Fig 3c in a revised version) were reported separately in Fig 2b (Fig 3b now). We show that in a range $d_{\text{bead}} = 0$ to 300 scatterers are too small to pin the roller vortex and instead get incorporated into the vortex structure. Once there, the number of scatterers influences the behavior of the vortex and we explore this scenario in Fig 2b (Fig 3b now).

Reply to Reviewer 3

We thank the Reviewer for a careful reading of our work and comments aimed to improve the clarity of our presentation. We are glad that the Reviewer finds our work fascinating and would be pleased to see it published in Nature Communications. The Reviewer made a number of comments and suggestions. In particular, the Reviewer suggested extending explanatory part of the paper that we fully addressed in the revised manuscript. Below are point-by-point replies to the Reviewer's comments:

1) *What sets the frequency at which vortices form? What is the relevant physics? For example, one might guess that particle motions arise through competition of magnetic and viscous torques at low Reynolds numbers. These two torques are of comparable magnitude on a particular time scale. How does that time scale compare to that of the applied frequency? On a related note, what is the magnetic moment of the particles? Perhaps these questions were addressed in previous work (ref. 26); however, it is necessary to repeat here for the benefit of the reader.*

We agree with the reviewer that short summary on the relevant physics of magnetic rollers that we introduced in our previous work would be beneficial for the overall clarity of the manuscript. The onset of rolling is attributed to the spontaneous symmetry breaking of clockwise/counterclockwise rotations of individual magnetic spheres in a uniaxial (vertical) alternating

magnetic field achieved when $\text{Im}(u[-p^2, 2q]) - p > 0$ (stability limit of the robust particle rotations) [29]. Here $u[x,y]$ is the Mathieu characteristic exponent function, $p = a_r/(\omega I)$, $q = \mu B_0 / (\omega^2 I)$, $a_r = 8\pi\eta R^3$ is the rotational drag coefficient, I is a momentum of particle inertia.

In the state of a steady rotation the magnetic torque on the particle ($T_m = \|\boldsymbol{\mu} \times \mathbf{B}\|$) is balanced by its viscous torque ($T_v \approx 8\pi\eta R^3 \omega$) and characteristic viscous time τ_v is comparable with a time scale of the applied field frequency τ_f . We added corresponding discussion to the revised manuscript. Relevant time scales (applied external field: $\tau_f \sim 1/f \approx 2.3 \cdot 10^{-2}$ sec; viscous time: $\tau_v \sim L^2/\eta$, here $L = 2 \cdot R_{Ni} \sim 140 \mu\text{m}$ is the characteristic size of the roller, $\tau_v \approx 2 \cdot 10^{-2}$ sec) and information on the magnetic moment of particles are now amended to the Methods section.

We also added extended explanatory part related to the physics behind the roller vortex formation on a flat surface (see also response to comment 1 of the Reviewer 2). The vortex state corresponds to the maximum correlation length between rollers and exists close to the boundary of the stability limit of the robust particle rotations mentioned above. For our system it is about 40Hz.

Corresponding discussions are now added to the revised text.

2) How do particles interact? Is it primarily magnetic dipole-dipole interactions or hydrodynamic interactions? How do these interactions facilitate vortex formation? Based on the “solid”-like character of the vortices, one might conjecture that repulsive dipolar interactions are significant.

Particles interact through induced hydrodynamic flows and dipole-dipole interactions. Both interactions are important and significant. The vortex structure emerges at a verge of a fine balance between these interactions. Each magnetic roller induces an anisotropic time-averaged interaction profile: rotation of the sphere in the fluid ($Re > 1$ for rollers, inertia is important) creates attractive hydrodynamic interactions in the lateral direction (along the axis of rotation, perpendicular to the direction of the roller motion) and repulsive in the direction perpendicular to the axis of rotation (along the direction of the roller motion) [38, 39]. As a result, rollers hydrodynamically attract neighbors laterally and repel them if they are along the rollers direction of motion. Both forces decay as $1/r^3$ [39, 40]. Time averaged magnetic interactions, on the other hand, are attractive along the rollers direction of motion and repulsive in lateral direction. Corresponding magnetic time averaged forces decay as $1/r^4$ [15]. Thus, hydrodynamic and magnetic interactions between rollers keep the roller vortex from falling apart (long range attractions prevent rollers from departing the vortex), or collapsing to clusters (short range repulsions keep rollers from getting too close to form chains or clusters). To further demonstrate the importance of the hydrodynamic interactions in the formation of roller vortices we eliminated the hydrodynamics and conducted similar experiments in air. We added a new Supplementary video demonstrating the state of the roller system in air (Supplementary Video 5). No vortices or flocks have been observed in a full range of the number densities and field parameters where steady rolling of particles was possible in the air.

The related discussions are now added in the text of the manuscript.

3) How does the speed of the vortex scale with its size? The Authors already have data on vortices of different sizes (Fig. 2a).

The vortex speed on a flat surface does not show dependence on the size and large vortices are approximately as mobile as small ones. A new supplementary Figure S2 has been added to illustrate the results.

We also added corresponding discussion to the manuscript.

4) What sets the characteristic size of the vortices (assuming there is a preferred size)?

A system of rollers at a flat surface can support a wide range of vortex sizes at a fixed experimental condition. The size of the vortex is selected by a dynamic self-induced densification of rollers and a range of different sizes can be realized in the system. We added a supplementary Figure S2 (panel b) demonstrating a histogram of the observed roller vortex sizes realized on a flat surface at the same experimental conditions. It is possible to have vortices as small as 1 mm and as large as 3.2 mm in diameter. Large vortices are less stable and may fall apart on average faster to form smaller entities; on the other hand small vortices may evaporate to a gas. On average the most probable size in the studied system was about 2 mm. This size could be altered by the number density of rollers.

We added corresponding comments in the revised manuscript.

Reviewer #1 (Remarks to the Author):

The author of the manuscript have addressed very carefully all my concern/queries. I'm thus happy to support publication of the article in Nature comm.

Reviewer #2 (Remarks to the Author):

The authors have satisfactorily addressed my queries and recommendations. I am happy to recommend publication.

Reviewer #3 (Remarks to the Author):

The Authors have made significant additions and revisions to the manuscript to address the Reviewer's questions and concerns. The changes help to clarify the physical mechanisms underlying vortex formation. I support publication in Nature Communications.

The manuscript (in particular the new additions) would benefit from further proofreading. For example, on page two there are several quantities that are introduced without definition (ω , m , η).

All three reviewers support publication of the paper in Nature Communications. We thank all reviewers for careful reading of our manuscript.

Reply to Reviewer 1

Reviewer 1 has no additional comments.

Reply to Reviewer 2

Reviewer 2 has no additional comments.

Reply to Reviewer 3

We thank the Reviewer for a careful reading of our work. The reviewer had a final comment related to proofreading:

The manuscript (in particular the new additions) would benefit from further proofreading. For example, on page two there are several quantities that are introduced without definition (ω , m , η).

We proofread the manuscript and added definitions for ω , m and η in the text.